# Closed-Loop Cavity Shear Layer Control Using Plasma Dielectric Barrier Discharge Actuators

Pavel N. Kazanskii 

Joint Institute for High Temperatures of the Russian Academy of Sciences (JIHT RAS), 125412 Moscow, Russia; fokkoo@yandex.ru

**Abstract:** The complex unsteady flow in cavities leads to the formation of large-scale disturbances in the shear layer. Natural closed-loop mechanisms provoke a dramatic increase in pressure pulsations and aerodynamic noise. This paper presents the experimental study of pressure fluctuations in closed-loop control in rectangular cavities using plasma dielectric barrier discharge. The flow velocity was 37 m/s, and the Reynolds number based on a cavity depth was approximately 120,000. The discharge ignition near the leading edge of the cavity provoked the shear layer restructuring. It was found that pressure fluctuations with an amplitude of 120 dB occur at frequencies 480 and 820 Hz. Frequency modulation of the discharge at resonant peaks was carried out by changing the phase shift of the power supply. The peak amplitude was reduced or increased by phase shifts from natural disturbances to forced ones. The optimum energy input was 50 W/m. This was three times less than the power consumption of the open-loop mode. The PIV visualization was organized in the phase-locked mode. The pressure spectrum corresponds to the magnitude of coherent structures in the shear layer of the cavity.

**Keywords:** plasma actuator; DBD; plasma aerodynamics; cavity; active control; boundary layer; shear layer; closed-loop control

## 1. Introduction

Cavity separation flow control has a significant impact on various scientific [1], military [2,3], gas transport [4], vehicle [5], environmental [6] and other tasks. The natural feedback present in this flow initiates self-oscillations at a set of frequencies called Rossiter modes [7]. As one moves, the interaction of coherent vortices present in the shear layer with the cavity trailing edge leads to the generation of acoustic waves that propagate upstream in the cavity recirculation region. Excitation of the instability wave at the leading edge by the acoustic waves closes a natural feedback loop. Active control of the cavity flow is based on the control of these main stages of tone generation.

The principal model developed by Rossister was improved in [8,9]. Improvements included accounting for the growing thickness of the shear layer and the finite depth of the cavity when determining the resonant frequencies. However, these models do not predict the amplitudes of the resonant modes. The earliest analysis of linear stability was carried out in [10]. The model also included an integral gain perturbation in the longitudinal direction. This made it possible to predict the dominant mode (n = 1, 2, or 3) of oscillations. In [11], the gains from the linear stability of the shear layer were used to predict the relative intensities of perturbations. These models have provided some guidelines for the development of active control systems. Modeling of large eddies in [12] demonstrated that the flow structure of two-dimensional caverns is strictly three-dimensional. The oscillation frequency can be predicted by the two-dimensional models quite well. On the other hand, the three-dimensionality needs to be taken into account for the estimation of the oscillation amplitudes.

A number of actuator types were used for cavity flow control. The influence of the flow structure of one cavity on another when they are located in tandem is considered in [13]. The dual-cavity structure is adaptable and effective in reducing losses under all operating conditions in the study. Among them are stationary [14] and non-stationary jets [15], piezo valves [16,17], liquid generators [18], synthetic jets [19], and plasma actuators.

Over the past decades, DBD discharges have become widespread in solving scientific and engineering problems. The high response time and the ability to be used on curved surfaces allowed this type of actuator to conquer its field. The most important DBD feature is the ability to sustain a large-volume discharge at atmospheric pressure without collapsing into a constricted arc. The low generated thrust [20] and efficiency [21] of such actuators generally limit the introduction of additional impulse into the flow. However, significant local heat generation made it possible to use DBD actuators in a wide range of tasks [22,23], including boundary layer flow control [24]. Dielectric barrier discharge was used to alter the shear layer geometry by introducing longitudinal vortices at low velocities [25]. Mode excitation was demonstrated by localized plasma filaments subsonic [26] and supersonic [27] conditions in terms of initiating nonlinear inter-mode interactions in the resonator.

The purpose of this work is to study the possibility of closed-loop control of a flow in a shallow cavity flow with a DBD plasma actuator. The real-time closed-loop system was designed to perform a stationary one-dimensional control of the zero azimuthal modes. It seemed important to introduce forced perturbations in the boundary layer in antiphase to natural stochastic perturbations, so that the sum of the amplitudes of both perturbations vanishes.

## 2. Materials and Methods

The experimental studies were carried out in the subsonic wind tunnel in JIHT RAS. The aerodynamic setup is shown in Figure 1a. The maximum flow velocity in the test section was up to 70 m/s. The test section had a cross section of $0.1 \times 0.1$ m and a length of 0.8 m, so the cavity width was W = 100 mm. It provided optical access to the cavity and general control of the experiment. The cavity was formed in the middle of a bump, formed by two wall inserts. The cavity depth was D = 50 mm, and the length was regulated in a wide range (L = 0–240 mm) by shifting the trailing part of the bump. A ceramic insert of 1 mm thickness with aluminum electrodes was installed flush with the streamlined surface of the leading edge of the cavity. The high-voltage electrode was 85 mm long and 0.05 mm thick. The distance between the electrode and the cavity leading edge was 5 mm. The ground electrode was installed under the ceramic layer and covered by a dielectric compound to avoid undesirable breakdown. The experiment was carried out under room air conditions.

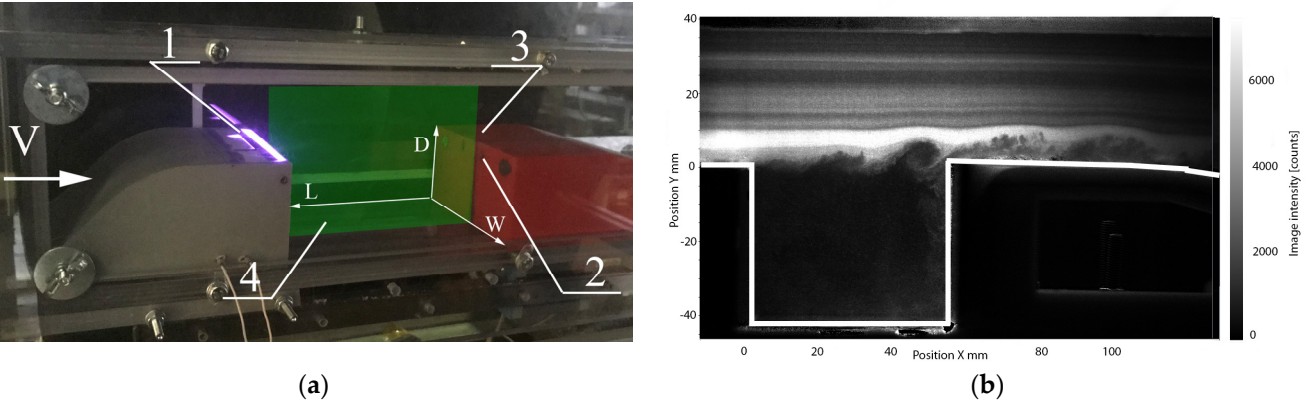

(**a**)                                                                (**b**)

**Figure 1.** Test section (**a**): 1—ceramic insert, 2—pressure scanner, 3—insert model of the diffuser, 4—laser knife. (**b**) Flow visualization by particle seeding for PIV method.

The oncoming flow velocity was measured using a Pitot tube with static ports on the side. The tube was mounted in the upper stream side of the test section at two calibers from the confuser. The pitot tube flow measurements were compared to the flow rate estimation of the fan power and channel hydrodynamic resistance. In addition, PIV data made it possible to obtain more accurate measurements using an independent method, as shown in Figure 1b.

Pressure fluctuations were measured using a miniature pressure sensor, Kulite XT-140 (M), with a pressure range of 1.7 bar and a self-resonant frequency of 240 KHz. The diameter of the sensor was 3 mm. The pressure sensor was installed at the trailing cavity wall 2 mm below the trailing edge. The signal from the sensor was filtered at 50 kHz and recorded by 12-bit ADC. The isolated power supply, shielded signal cables, and electronics were used to prevent electromagnetic interference between the DBD and the sensor.

The flow velocity was measured using the PIV LaVision FlowMaster system. The flow was seeded with ~1 μm oil particles. The dynamic relaxation time was 1–2 μs. The particles were illuminated by two successive laser pulses. The images were recorded using a 4 Mpix camera. Image processing was carried out using the cross-correlation method with a 32 × 32 pix window size and 50% overlap. The resulting resolution of the vector fields was 0.5 mm in the plane of the laser knife. The resulting velocity field was obtained by averaging 150 instantaneous frames.

Control feedback was implemented on the LCard E440ADC-DSP-DAC module. Signal processing in DSP included FIR filtering of the pressure sensor signal. Bandpass filters with a typical bandwidth of 125 Hz were used. The filter was centered on a desired Rossiter mode, and then an adjustable delay was introduced with respect to the pressure sensor readings. For the independent control of pressure pulsation amplitudes at different frequencies, it was possible to turn on the bins corresponding to their bandwidth. The output signal was used to modulate the switching frequency at the resonant power supply of the DBD actuator. Thus, the device had sufficient bandwidth that was capable of producing a control input at any instant in time consisting of several frequencies, each at its own amplitude and phase. The actuator had a time response commensurate with the time scales of the cavity flow dynamics. It also triggered the PIV system, which makes it possible to carry out a phase-locked measurement of the perturbation development under resonant conditions, Figure 2. The flow field visualization resolution was 1 mm.

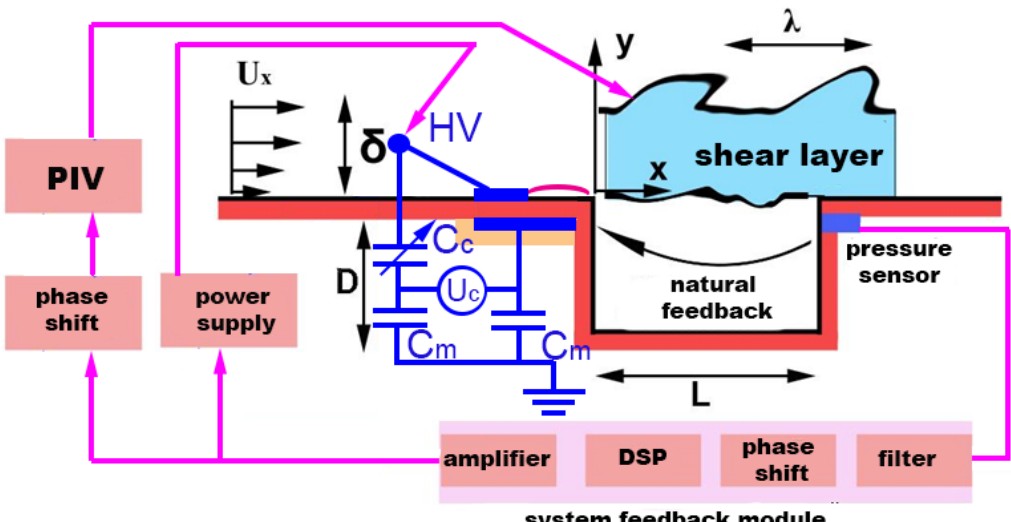

**Figure 2.** Schematic diagram of pressure pulsation control using a closed-loop system.

An AWG-4082 generator was used to form the master pulses. Frequency modulation of the signal was implemented to control the power actuator. Discharge was operated at approximate frequency $f_{gen}$ = 127 kHz, and typical voltage amplitude less than 8 kV. The

flow control in an open loop system was held in [25]. In the current paper, the closed-loop control was organized (see Appendix A).

The method of volt–coulomb cycloramas was used to measure and control the discharge power. It consists of analyzing the DBD charge transferred cyclic dependency on the instantaneous value of the electrode voltage [28,29]. The measurement diagram is shown in Figure 2. The buried electrode was connected to the measuring capacitance $C_m$ with a nominal value of 1 nF. According to the current continuity equation, the charge flowing through the discharge cell is equal to the charge of the capacitance $C_m$. An analog compensation circuit in the form of a capacitor bridge was used in order to exclude parasitic capacitive current through the electrode system from measurements. A tunable vacuum capacitor $Cc$ = 3–50 pF was installed in its upper arm. The capacity of this capacitor was set equal to the capacity of the electrode system in the absence of a discharge. With this scheme, the power dissipated in the discharge can be calculated as

$$P = fE_T = fC_m \oint U_c(t)dU(t) \, , \tag{1}$$

$f$—applied voltage carrier frequency, $E_T$—energy input into the discharge during the period of supply voltage, and $U_c(t)$—voltage difference across measuring capacitors $C_m$. $U_c(t)$ was measured using a Pintek DP-150 differential voltage probe (5% accuracy, 150 MHz bandwidth). $E_T$ remained virtually unchanged in the frequency range corresponding to the depth of discharge modulation. The power measuring error was estimated at 10% [30].

## 3. Results and Discussion

The real flow was highly unsteady and complex, so the mean flow parameters do not necessarily correspond to the instantaneous values at any instant time. The transition from one dominant cavity mode to another can be caused by a change in the geometry L/D and flow velocity. For certain geometries and velocities, the DBD discharge could initiate a restructuring of the flow regime from one mode to another. However, such modes were not interesting for consideration and were deliberately avoided in the study. The main results were obtained with a two-dimensional cavity length of 61.5 mm. The oncoming flow velocity was 37 m/s. The Reynolds number based on a cavity depth of 50 mm was approximately 120,000. The average flow field velocity is shown in Figure 3. One can see the typical flow structure in a rectangular cavity. The walls of the cavity coincide with the wall of the test section. Thus, the air cannot be drawn continuously into the eddies from the external stream and escape in a trailing vortex system shed from the cavity. The thickening of the shear layer increases from a few mm near the leading edge up to 10 mm near the trailing edge of the cavity. So, the flow separates from the front edge and does not reattach along the roof of the cavity.

The general spectrum of pressure fluctuations is shown in Figure 4. The natural resonance oscillations were found at frequencies 480, 820, and 1325 Hz, which corresponded to the Strouhal numbers 0.57, 1.1, and 1.8, respectively. The latter frequency dominates in the cavity flow. At first, the closed-loop filter was selectively tuned on all three detected tones separately. The filter bandwidth was 127 Hz. The influence of the plasma actuator could reduce the peak amplitude from 120 to 112 dB with a signal phase shift of 4.2 rad or increase it to 127 dB if the phase shift is 6 rad. The narrow-band resonant peak at a frequency of 1300 Hz was changed within the measurement error. It should be noted that the influence of the plasma actuator on resonances picks at 480 and 820 Hz frequencies occur independently of each other. All four combinations of influence on resonant peaks were possible: 1. increase of both resonances, 2. decrease of both resonances, 3. increase of the first resonance and decrease of the second resonance, and 4. reverse case.

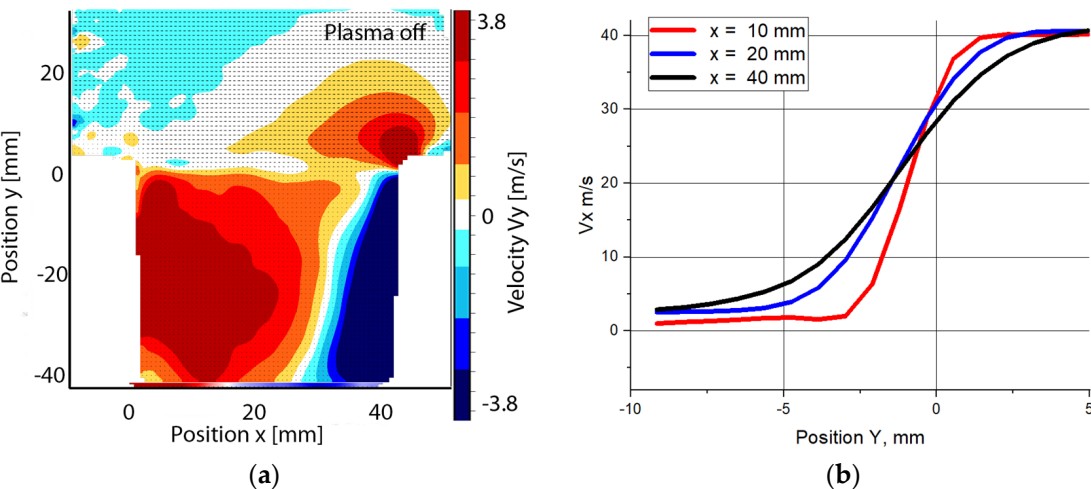

(**a**)

(**b**)

**Figure 3.** The average flow field velocity while the actuator is switched off (**a**). Velocity $V_x$ distribution in the cavity in different sections (**b**).

(**a**)

(**b**)

(**c**)

(**d**)

**Figure 4.** The spectrum of pressure pulsations in the absence of the actuator, in the buildup modes f = 6.0 rad, and suppression f = 4.2 rad perturbations. The general spectrum (**a**), and the spectrum at: 500 Hz (**b**), 820 Hz (**c**), 1320 Hz (**d**).

The maximum resonant peak amplitude due to the plasma phase shift is shown in Figure 5. It can be seen that the experimental points can be approximated by a harmonic function as follows:

$$SPL_{\text{plasma on}} = A \cos(\varphi) + SPL_{\text{plasma off}} \qquad (2)$$

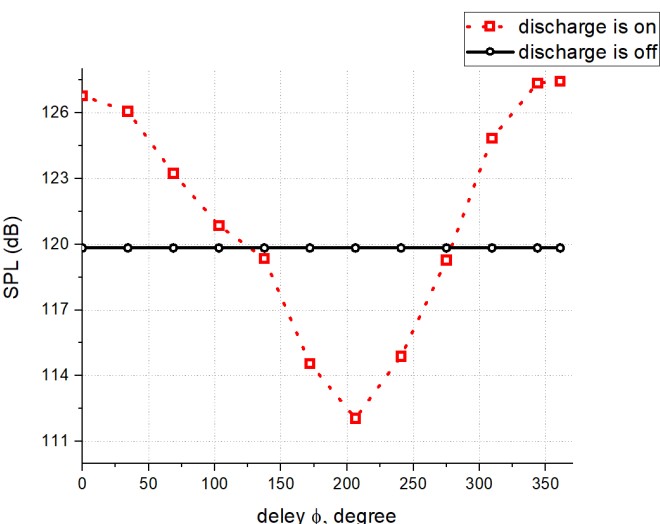

**Figure 5.** Maximum amplitude versus phase shift.

It should be noted that the change in the pressure amplitude in the buildup and suppression modes was the same (~7 dB), including at other velocities and cavity geometries. Apparently, this is due to additional nonlinear mechanisms that limit the growth and suppression of perturbations, respectively. A slight change in the phase shift will result in a slight change in the amplitude of the resonant peak. Thus, based on the operation of a plasma actuator with feedback, a good quality of control can be ensured.

Additional studies of the maximum pressure amplitude of the resonant peak at a frequency of 480 Hz were carried out. The phase shift was 4.2 rad and was optimal for suppressing resonant oscillations at this frequency. During the experiment, the voltage of the autotransformer changed, as shown in Figure A1. So, the actuator energy input was controlled. It was found that with an increase in the energy input from 20 to 50 W/m, the feedback system operates in a quasi-linear mode, and the resonance peak decreases Figure 6. However, as the voltage at the high-voltage electrode increases, the decrease in the resonant peak begins to level out. Moreover, at an energy input of 75 W/m, the plasma does not reduce pressure fluctuations at this frequency, and with further growth, the resonance peak builds up. Apparently, at an energy input of 75 W/m, the amplitude of induced pressure pulsations in the boundary layer is twice as high as the amplitude of natural pulsations in antiphase. Thus, a further increase in the energy input leads to the fact that the development of perturbations in the wake of the cavity occurs due to forced oscillations introduced by the plasma actuator.

Visualization of the shear layer flow structure in the cavity is shown in Figure 7. The vertical component of the flow velocity is highlighted with a discrete scale. The velocity fields were obtained by averaging 150 instantaneous frames with the same phase delay relative to the pressure sensor reference signal. The left picture shows the vortex structure while the discharge is switched off. The vertical velocity is in the range of −3.6 m/s in the area, which is marked in a circle (blue color). The center picture shows the DBD plasma actuator operation in disturbance suppression mode. One can see the decries of flow to −1.9 m/s as the marked aria colors in «ice blue». The major difference can be seen in the comparison of the suppression mode and the swing mode. The maximum flow velocity comes to −3.8 m/s, and the color of the picked aria goes to the navy color. The disturbance wavelength λ can be measured while observing the evolution of the vortex wave motion. It is 15 mm near the leading edge. As one moves downstream, the disturbance wavelength

increases rapidly up to 20 mm, so that time average length is 18 mm. As it is known [10], during any particular mode of oscillation, the wavelength of the disturbances bears nearly a constant relationship to the width L of the cavity. The spacing of the periodic vortices, which are shed from the downstream corner at the frequency of cavity oscillations, further confirms such an integral relationship between L and λ. The present result suggests the following:

$$L = \lambda\,(N + 0.5), \tag{3}$$

where N is an integer. Thus, the N + 1 (second) mode is shown. One can see the flow field while the actuator is switched off; the actuator operates in the suppressing mode, and when the actuator operates in the excitation mode. It is clearly seen that the main flow structure changes lie in the region of the shear layer. The high-velocity region flow structure outside the cavity and the flow structure inside the cavity changed insignificantly.

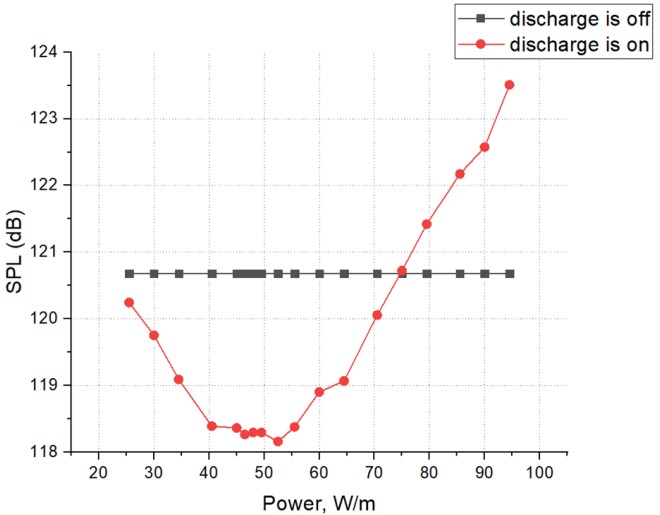

**Figure 6.** The dependence of the amplitude of the resonant peak on the power.

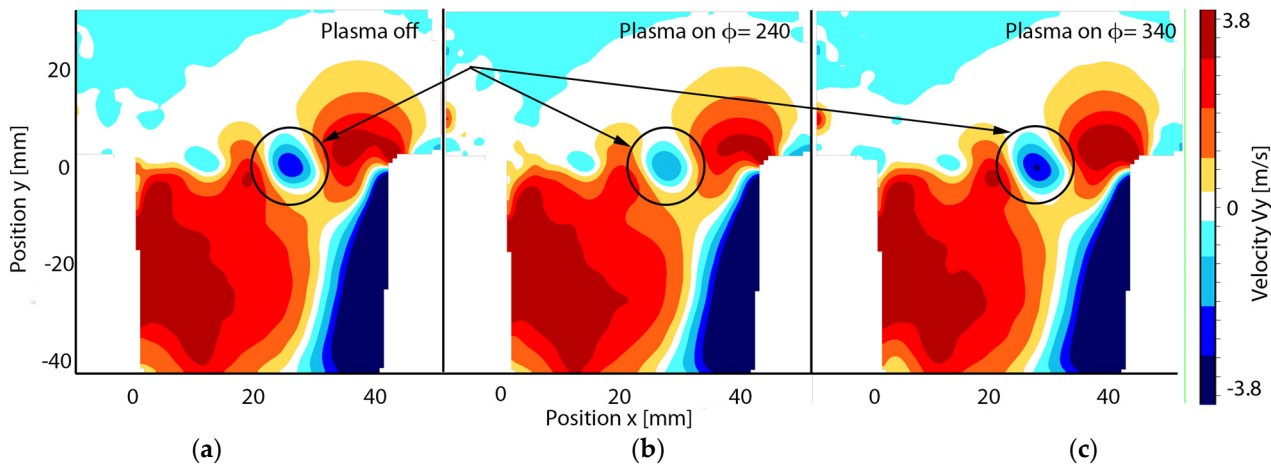

**Figure 7.** Comparison of flow field patterns without discharge and with phase delays for buildup and suppression of disturbances. (**a**) discharge is off, (**b**) discharge is on φ = 240, suppression mode, and (**c**) discharge is on φ = 340, swing mode.

The average vertical velocity component in the shear layer is shown in Figure 8a. It can be seen that as the fluid moves downstream, the velocity amplitude increases 1.3, 1.6, and 1.7 times with each period for the suppression, unperturbed, and buildup modes, respectively. The evolution of the velocity amplitude in the shear layer is shown in Figure 8b. It can be approximated linearly. However, the rate of increase in the amplitude for all

three presented cases is different and, accordingly, 1.3 times higher in the excitation mode compared to the suppression one. It can also be seen that the amplitude pulsation does not saturate and apparently continues to grow in a certain area downstream of the trailing edge.

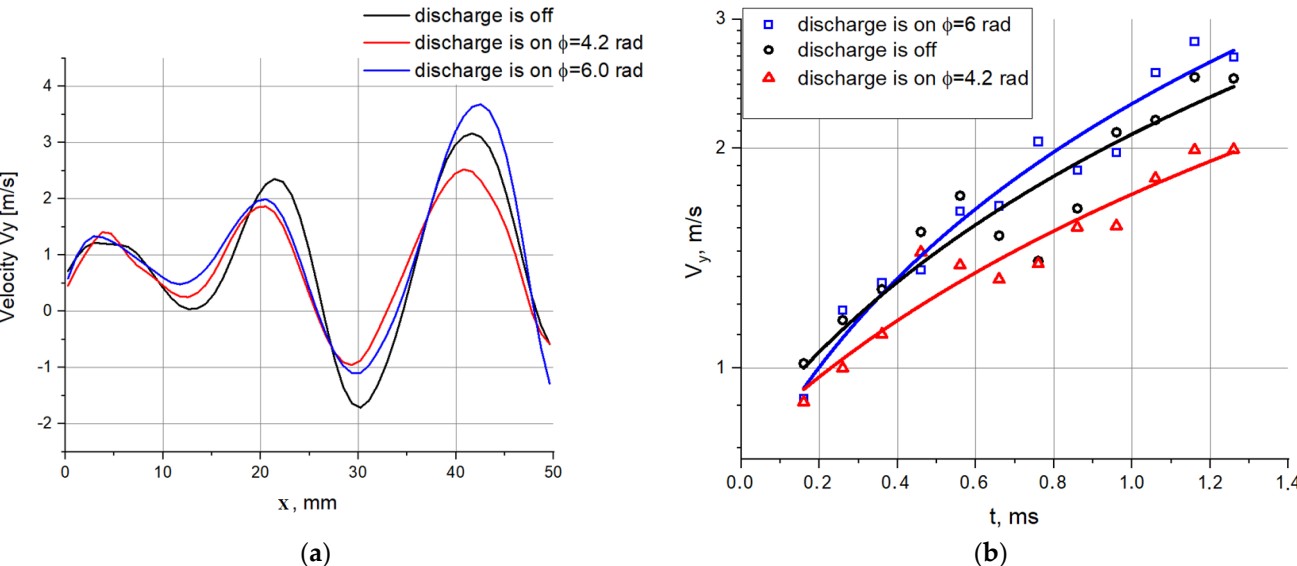

(a)  (b)

**Figure 8.** (**a**) The vertical velocity component for an unexcited flow, as well as for the case of buildup and suppression of disturbances. (**b**) The vertical velocity component for an unexcited flow, as well as for the case of buildup and suppression of disturbances.

## 4. Conclusions

The DBD actuator near the leading edge of the cavity was studied at a flow velocity of 37 m/s, which corresponded to the Reynolds number based on a cavity depth of approximately 120,000. The closed-loop flow control by means of a pressure sensor signal was carried out. It was possible to influence the natural feedback circuit and reduce pressure fluctuations in the cavity. The discharge was ignited in antiphase to natural perturbations. The selection of the amplitude coefficient suppressed the initial disturbances near the leading edge of the cavity by 8 dB. The vortex evolution visualization showed that the growth rate was changed at the discharge ignition. It should be noted that in the antiphase mode, there is an optimal amplitude coefficient of forced disturbances. A further increase in the amplitude coefficient only leads to an increase in the pressure fluctuation in the cavity.

The discharge ignition in phase mode revolved around the influence. The buildup mode was observed. The pressure pulsations resonance pick amplitude was increased by 7 dB. However, the maximum resonance pressure amplitude is limited by other non-linear mechanisms that prevent the growth of the resonant pressure peak. These mechanisms need further research. The closed-loop cavity flow control seems to be promising firstly in order to optimize energy input.

The impact amplitude spectrum analysis shows that more than 70% of disturbances fall on pressure fluctuations in the range from 0.2 to 0.6 of the maximum amplitude. Perturbations less than 0.2 and more than 0.6 of the maximum amplitude account for equal shares of 15%. Operating the actuator at a significantly lower "average" voltage leads to a significant reduction in power consumption from 130 W/m in an open-loop regime to 50 W/m in a closed-loop regime. This reduces the heating of structural elements and increases the reliability of the system since the discharge does not work all the time near pre-breakdown voltages for ceramics and structural components.

**Funding:** The work was supported by the Russian Science Foundation grant No. 22-29-00353.

**Data Availability Statement:** The source data .vc7 of Figure 7 are available in a publicly accessible repository https://drive.google.com/file/d/1fj7bb4-j1HzLQh8uKB6Kez3WAoOVp8SV/view?usp= sharing (accessed on 10 October 2023). The additional data presented in this study are available on request from the corresponding author.

**Acknowledgments:** The author expresses acknowledgment to Moralev I.A. for the technical support, physics discussion of the cavity flow processes, and editing of the paper.

**Conflicts of Interest:** The authors declare no conflict of interest. The funders had no role in the design of the study; in the collection, analyses, or interpretation of data; in the writing of the manuscript; or in the decision to publish the results.

**Appendix A**

An AWG-4082 generator was used to form the master pulses. The main discharge frequency $f_{gen}$ = 127 kHz was set higher than the resonant frequency $f_{res}$ = 122 kHz Figure A1. The frequency discharge modulation was tuned to the natural frequency of cavity pressure oscillations. In the first stage, the flow control was carried out in an open loop system. The internal AWG-4082 generator sinusoidal signal with a fixed voltage amplitude $U_{CL\_off}$ = 5 V was used as a signal of the shift modulation frequency. In this case, the carrier frequency shift was $df_{car}$ = 5 kHz, so that

$$f_{car} = f_{gen} - df_{car} = f_{res} \tag{A1}$$

The voltage amplitude $U_m$ was correspondingly constant and reached up to 8 kV.

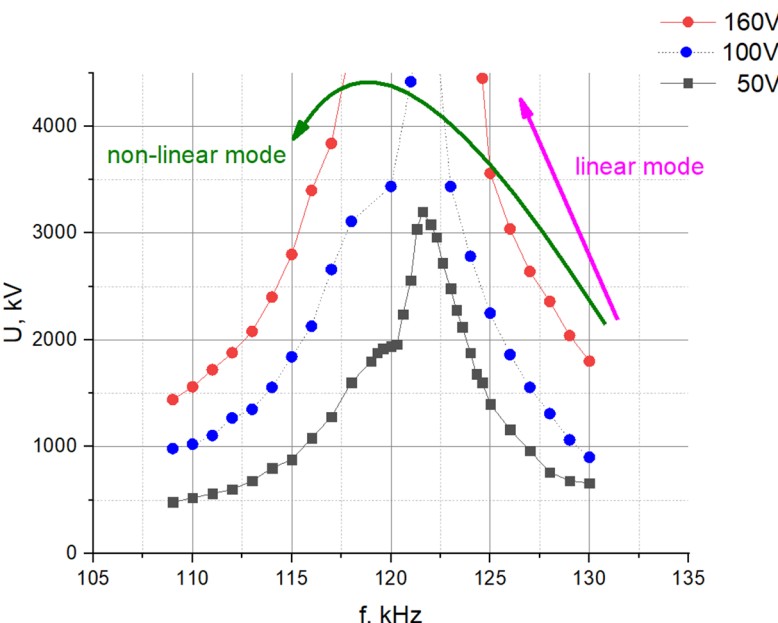

**Figure A1.** Dependence of the discharge voltage on the carrier frequency. The input voltage on the amplifier was set by a variable autotransformer.

In the second stage of the study, the case of the closed-loop mode was realized. The sinusoidal signal of the generator was replaced by a control signal. The carrier frequency shift $df_{car}$ now was proportional to the amplitude of the pressure sensor signal at the resonant frequency. If the pressure sensor amplitude was 80% of the maximum, the modulation signal voltage corresponded to 3 V, and the discharge voltage was 8 kV. Thus, the DBD discharge was mostly ignited in the generator linear mode. This area is circled by a purple rectangle. Figure A2 shows the amplitude distribution of pressure pulsation pulses in the shear layer of the cavity at three different frequencies of resonant peaks, and 99% of the perturbations corresponding to the linear mode of the generator, as well as 1%

of rare perturbations, corresponding to the nonlinear mode, lead to conditionally incorrect operation of the system. The influence of such perturbations was considered insignificant.

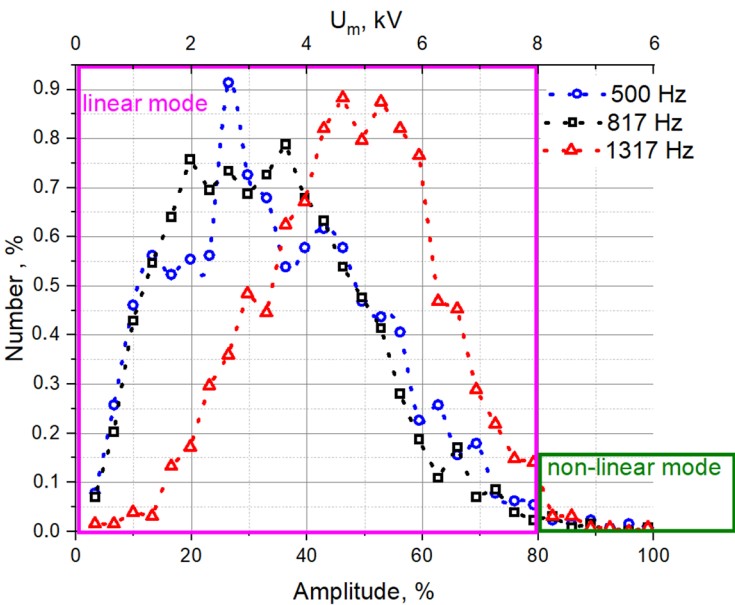

**Figure A2.** Voltage amplitude pulsation histogram for three resonant peaks.

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
