# Peer review of "Closed-Loop Cavity Shear Layer Control Using Plasma Dielectric Barrier Discharge Actuators"

_aerospace, doi:10.3390/aerospace10100888_

Round 1

Reviewer 1 Report

Review on the manuscript intitled “Close loop cavity shear layer control by means of plasma dielectric barrier discharge actuator”, by Kazanskii.

The manuscript presents an experimental work that demonstrates the feasibility of plasma actuators for close loop cavity shear layer control. The topic is interesting, the article is well written and structured, presenting several results with appropriate discussion. However, in my opinion, some improvements should be made to achieve the quality standards of the journal of “Aerospace”. Please consider my following comments:

1-     In my opinion the abstract should be improved and rearranged. It presents too many technical information that in my opinion does not need to be mentioned here, and it lacks other important information. I think it should include a a brief introduction to the topic and why it is worth of investigation. Then, the indication of the research methods and approach which, in my opinion, can be condensed and does not need all the technical details that are provided. And, to finalize, one or two key conclusions achieved with the current work.

2-     The introduction is well written and emphasizes well the importance of cavity separation control, however, in my opinion it should provide more information about plasma actuators, since it only presents two sentences about these devices. Authors should explain better the features of these devices and why they are appealing for flow control applications, briefly explain their operation and mention also that they have possible application not only on the flow control field but also within the heat transfer field. You may consider the following works about plasma actuators:

a.       https://doi.org/10.1007/s00348-014-1846-x

b.      https://doi.org/10.1016/j.sna.2022.113391

c.      https://doi.org/10.1016/j.expthermflusci.2023.110950

3-     In the materials and methods section authors should specify the dimensions of the wind tunnel, the dimensions of the model and present the blockage ratio on the test section.

4-     More information should be added about the PIV system, including the model of the system, or model of the camera, laser, software used for cross-correlation analysis, type of particles used for the seeding etc.

5-     In Figure 3 a subfigure is missing, it should have three subfigures but only two are shown.

6-     In Figure 4, the subfigures should be named as a), b), c) and d), as done in Figure 3. In addition, author should explain the reason for the selection of the frequency ranges of subfigures b), c) and d). Please comment why the effect of the plasma actuation is different in the peaks found for frequencies higher than 1350Hz.

The manuscript should be proofread, just very minor language mistakes need to be corrected.

Author Response

Dear reviewer. Thank you for your critical assessment of the paper on substantive issues. Your comments will truly improve readers' understanding of the text and enhance the work.

Reviewer 2 Report

The author describes phenomena related to the impact of DBD discharges on the flow of air streams in a cavity test.

1) The quality of drawing 1 and the PIV image looks as if it was pasted from other tests. Please provide a better quality photo and mark the individual dimensions in drawing 1, which distances are adjustable and what the individual dimensions are.

2) No PIV drawings showing visual changes in individual configurations/layout changes.

3) The work is written in a chaotic manner, the numbering of subsequent chapters is incorrect (point 4 is missing)

4) No photos from the flow without the DBD system turned on and with the DBD system turned on to illustrate visual changes

5) Figure 3 is missing one figure (there are a, b, c) in the publication on page 4, figure 3 consists of only 2 graphs.

6) There is no description of the program used to analyze Figure 3a), Figure 7.

7) On what basis was the power of BDB discharges calculated? There is no power measurement scheme and it is not indicated on what basis the power was calculated and for what frequency range it was calculated. As we know, the effect of frequency has a large impact on the power of discharges.

8) The title of the publication does not reflect the content of the work. The work describes a method of influencing changes in pressure in the cavity using DBD discharges, not on the laminarity of the flow in the boundary layer, as is the case, for example, on the wing surface, where the detached air streams are "glued" back to the wing surface, which allows for an increase, e.g. lift coefficient Cl.

Readers should receive understandable content and data that do not require the reader to guess how the experiment was carried out and the results obtained.

Minor mistakes

Author Response

(The authors gave the same response as above.)
